# Efficient Wasserstein Natural Gradients for Reinforcement Learning

**Ted Moskovitz**[*1]**, Michael Arbel**[*1]**, Ferenc Huszar**[1,2] **& Arthur Gretton**[1]
[1]Gatsby Unit, UCL     [2]University of Cambridge

## Abstract

A novel optimization approach is proposed for application to policy gradient methods and evolution strategies for reinforcement learning (RL). The procedure uses a computationally efficient *Wasserstein natural gradient* (WNG) descent that takes advantage of the geometry induced by a Wasserstein penalty to speed optimization. This method follows the recent theme in RL of including a divergence penalty in the objective to establish a trust region. Experiments on challenging tasks demonstrate improvements in both computational cost and performance over advanced baselines.

## 1 Introduction

Defining efficient optimization algorithms for reinforcement learning (RL) that are able to leverage a meaningful measure of similarity between policies is a longstanding and challenging problem (Lee & Popović, 2010; Meyerson et al., 2016; Conti et al., 2018b). Many such works rely on similarity measures such as the Kullback-Leibler (KL) divergence (Kullback & Leibler, 1951) to define procedures for updating the policy of an agent as it interacts with the environment. These are generally motivated by the need to maintain a small variation in the KL between successive updates in an off-policy context to control the variance of the importance weights used in fthe estimation of the gradient. This includes work by Kakade (2002) and Schulman et al. (2015), who propose to use the *Fisher Natural Gradient* (Amari, 1997) as a way to update policies, using *local* geometric information to allow larger steps in directions where policies vary less; and the work of Schulman et al. (2017), which relies on a *global* measure of proximity using a soft KL penalty to the objective. While those methods achieve impressive performance, and the choice of the KL is well-motivated, one can still ask if it is possible to include information about the behavior of policies when measuring similarity, and whether this could lead to more efficient algorithms. Pacchiano et al. (2019) provide a first insight into this question, representing policies using *behavioral distributions* which incorporate information about the outcome of the policies in the environment. The Wasserstein Distance (WD) (Villani, 2016) between those behavioral distributions is then used as a similarity measure between their corresponding policies. They further propose to use such behavioral similarity as a *global* soft penalty to the total objective. Hence, like the KL penalty, proximity between policies is measured globally, and does not necessarily exploit the local geometry defined by the behavioral embeddings.

In this work, we show that substantial improvements can be achieved by taking into account the *local* behavior of policies. We introduce new, efficient optimization methods for RL that incorporate the local geometry defined by the behavioral distributions for both policy gradient (PG) and evolution strategies (ES) approaches. Our main contributions are as follows:

**1-** We leverage recent work in (Li & Montufar, 2018a;b; Li, 2018; Li & Zhao, 2019; Chen & Li, 2018) which introduces the notion of the Wasserstein Information Matrix to define a *local behavioral similarity* measure between policies. This allows us to identify the Wasserstein Natural Gradient (WNG) as a key ingredient for optimization methods that rely on the local behavior of policies. To enable efficient estimation of WNG, we build on the recent work of Arbel et al. (2020), and further extend it to cases where the re-parameterization trick is not applicable, but only the score function of the model is available.

---

[*]Denotes equal contribution. Correspondence: `ted@gatsby.ucl.ac.uk`

**2-** This allows us to introduce two novel methods: *Wasserstein natural policy gradients* (WNPG) and *Wasserstein natural evolution strategies* (WNES) which use the local behavioral structure of policies through WNG and can be easily incorporated into standard RL optimization routines. When combined in addition with a global behavioral similarity such as a WD penalty, we show substantial improvement over using the penalty alone without access to local information. We find that such WNG-based methods are especially useful on tasks in which initial progress is difficult.

**3-** Finally, we demonstrate, to our knowledge, the first in-depth comparative analysis of the FNG and WNG, highlighting a clear interpretable advantage of using WNG over FNG on tasks where the optimal solution is deterministic. This scenario arises frequently in ES and in policy optimization for MDPs (Puterman, 2010). This suggests that WNG could be a powerful tool for this class of problems, especially when reaching accurate solutions quickly is crucial.

In Section 2, we present a brief review of policy gradient approaches and the role of divergence measures as regularization penalties. In Section 3 we introduce the WNG and detail its relationship with the FNG and the use of Wasserstein penalties, and in Section 4 we derive practical algorithms for applying the WNG to PG and ES. Section 5 contains our empirical results.

## 2 BACKGROUND

**Policy Gradient (PG)** methods directly parametrize a policy $\pi_\theta$, optimizing the parameter $\theta$ using stochastic gradient ascent on the expected total discounted reward $\mathcal{R}(\theta)$. An estimate $\hat{g}_k$ of the gradient of $\mathcal{R}(\theta)$ at $\theta_k$ can be computed by differentiating a surrogate objective $\mathcal{L}_\theta$ which often comes in two flavors, depending on whether training is *on-policy* (left) or *off-policy* (right):

$$\mathcal{L}(\theta) = \hat{\mathbb{E}}\left[\log \pi_\theta(a_t|s_t)\hat{A}_t\right], \qquad \text{or} \qquad \mathcal{L}(\theta) = \hat{\mathbb{E}}\left[\frac{\pi_\theta(a_t|s_t)}{\pi_{\theta_k}(a_t|s_t)}\hat{A}_t\right]. \tag{1}$$

The expectation $\hat{\mathbb{E}}$ is an empirical average over $N$ trajectories $\tau_i = (s_1^i, a_1^i, r_1^i, ..., s_T^i, a_T^i, r_T^i)$ of state-action-rewards obtained by simulating from the environment using $\pi_{\theta_k}$. The scalar $\hat{A}_t$ is an estimator of the advantage function and can be computed, for instance, using

$$\hat{A}_t = r_t + \gamma V(s_{t+1}) - V(s_t) \tag{2}$$

where $\gamma \in [0, 1)$ is a discount factor and $V$ is the value function often learned as a parametric function via temporal difference learning (Sutton & Barto, 2018). Reusing trajectories can reduce the computational cost at the expense of increased variance of the gradient estimator (Schulman et al., 2017). Indeed, performing multiple policy updates while using trajectories from an older policy $\pi_{\theta_{old}}$ means that the current policy $\pi_\theta$ can drift away from the older policy. On the other hand, the objective is obtained as an expectation under $\pi_\theta$ for which fresh trajectories are not available. Instead, the objective is estimated using importance sampling (by re-weighting the old trajectories according to importance weights $\pi_\theta/\pi_{\theta_{old}}$). When $\pi_\theta$ is too far from $\pi_{\theta_{old}}$, the importance weight can have a large variance. This can lead to a drastic degradation of performance if done naïvely (Schulman et al., 2017). KL-based policy optimization (PO) aims at addressing these limitations.

**KL-based PO methods** ensure that the policy does not change substantially between successive updates, where change is measured by the KL divergence between the resulting action distributions. The general idea is to add either a hard KL constraint, as in TRPO (Schulman et al., 2015), or a soft constraint, as in PPO (Schulman et al., 2017), to encourage proximity between policies. In the first case, TRPO recovers the FNG with a step-size further adjusted using line-search to enforce the hard constraint. The FNG permits larger steps in directions where policy changes the least, thus reducing the number of updates required for optimization. In the second case, the soft constraint leads to an objective of the form:

$$\text{maximize}_\theta \ \mathcal{L}(\theta) - \beta\hat{\mathbb{E}}\left[\text{KL}(\pi_{\theta_k}(\cdot|s_t), \pi_\theta(\cdot|s_t))\right]. \tag{3}$$

The KL penalty prevents the updates from deviating too far from the current policy $\pi_{\theta_k}$, thereby controlling the variance of the gradient estimator. This allows making multiple steps with the same simulated trajectories without degradation of performance. While both methods take into account the proximity between policies as measured using the KL, they do not take into account the *behavior* of such policies in the environment. Exploiting such information can greatly improve performance.

**Behavior-Guided Policy Optimization.** Motivated by the idea that policies can differ substantially as measured by their KL divergence but still *behave* similarly in the environment, Pacchiano et al. (2019) recently proposed to use a notion of proximity in *behavior* between policies for PO. Exploiting similarity in behavior during optimization allows to take larger steps in directions where policies behave similarly despite having a large KL divergence. To capture a sense of global behavior, they define a *behavioral embedding map* (BEM) $\Phi$ that maps every trajectory $\tau$ to a behavior variable $X = \Phi(\tau)$ belonging to some embedding space $\mathcal{E}$. The behavior variable $X$ provides a simple yet meaningful representation of each the trajectory $\tau$. As a random variable, $X$ is distributed according to a distribution $q_\theta$, called the *behavior distribution*. Examples of $\Phi$ include simply returning the final state of a trajectory ($\Phi(\tau) = s_T$) or its concatenated actions ($\Phi(\tau) = [a_0, \ldots, a_T]$). Proximity between two policies $\pi_\theta$ and $\pi_{\theta'}$ is then measured using the Wasserstein distance between their *behavior distributions* $q_\theta$ and $q_{\theta'}$. Although, the KL could also be used in some cases, the Wasserstein distance has the advantage of being well-defined even for distributions with non-overlapping support, therefore allowing more freedom in choosing the embedding $\Phi$ (see Section 3.1). This leads to a penalized objective that regulates behavioral proximity:

$$\text{maximize}_\theta \ \mathcal{L}(\theta) - \frac{\beta}{2} W_2(q_{\theta_k}, q_\theta), \tag{4}$$

where $\beta \in \mathbb{R}$ is a hyper-parameter controlling the strength of the regularization. To compute the penalty, Pacchiano et al. (2019) use an iterative method from Genevay et al. (2016). This procedure is highly accurate when the Wasserstein distance changes slowly between successive updates, as ensured when $\beta$ is large. At the same time, larger values for $\beta$ also mean that the policy is updated using smaller steps, which can impede convergence. An optimal trade-off between the rate of convergence and the precision of the estimated Wasserstein distance can be achieved using an adaptive choice of $\beta$ as done in the case of PPO Schulman et al. (2017). For a finite value of $\beta$, the penalty accounts for *global* proximity in behavior and doesn't explicitly exploit the local geometry induced by the BEM, which can further improve convergence. We introduce an efficient method that explicitly exploits the local geometry induced by the BEM through the Wasserstein Natural gradient (WNG), leading to gains in performance at a reduced computational cost. When global proximity is important to the task, we show that using the Wasserstein penalty in Equation (4) and optimizing it using the WNG yields more efficient updates, thus converging faster than simply optimizing Equation (4) using standard gradients.

## 3 THE WASSERSTEIN NATURAL GRADIENT

The Wasserstein natural gradient (WNG) (Li & Montufar, 2018a;b) corresponds to the steepest-ascent direction of an objective within a trust region defined by the local behavior of the Wasserstein-2 distance ($W_2$). The $W_2$ between two nearby densities $q_\theta$ and $q_{\theta+u}$ can be approximated by computing the average cost of moving every sample $X$ from $q_\theta$ to a new sample $X'$ **approximately** distributed according to $q_{\theta+u}$ using an **optimal** vector field of the form $\nabla_x f_u(x)$ so that $X' = X + \nabla_x f_u(X)$ (see Figure 6). Optimality of $\nabla_x f_u$ is defined as a trade-off between accurately moving mass from $q_\theta$ to $q_{\theta+u}$ and reducing the transport cost measured by the average squared norm of $\nabla_x f_u$

$$\sup_{f_u} \ \nabla_\theta \mathbb{E}_{q_\theta} \left[ f_u(X) \right]^\top u - \frac{1}{2} \mathbb{E}_{q_\theta} \left[ \| \nabla_x f_u(X) \|^2 \right], \tag{5}$$

where the optimization is over a suitable set of smooth real valued functions on $\mathcal{E}$. Hence, the optimal function $f_u$ solving Equation (5) defines the **optimal** vector field $\nabla_x f_u(x)$. Proposition 1 makes this intuition more precise and defines the Wasserstein Information Matrix.

**Proposition 1 (Adapted from Defintion 3 Li & Zhao (2019))** *The second-order Taylor expansion of $W_2$ between two nearby parametric probability distributions $q_\theta$ and $q_{\theta+u}$ is given by*

$$W_2^2(q_\theta, q_{\theta+u}) = u^\top G(\theta) u + o(\|u\|^2) \tag{6}$$

*where $G(\theta)$ is the Wasserstein Information Matrix (WIM), with components in a basis $(e_1, ..., e_p)$*

$$G_{j,j'}(\theta) = \mathbb{E}_{q_\theta} \left[ \nabla_x f_j(X)^\top \nabla_x f_{j'}(X) \right]. \tag{7}$$

*The functions $f_j$ solve Equation (5) with $u$ chosen as $e_j$. Moreover, for any given $u$, the solution $f_u$ to Equation (5) satisfies $\mathbb{E}_\theta[\| \nabla_x f_u(X) \|^2] = u^\top G(\theta) u$.*

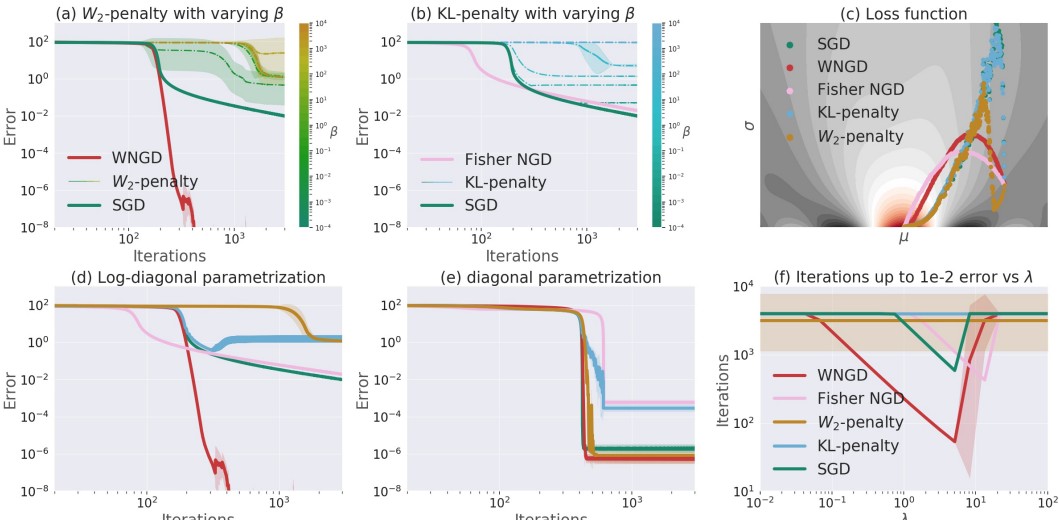

Figure 1: Different optimization methods using an objective $\mathcal{L}(\theta) = \mathbb{E}_{q_\theta}[\psi(x)]$ where $q_\theta$ is a gaussian of 100 dimensions with parameters $\theta = (\boldsymbol{\mu}, v)$. Here $\boldsymbol{\mu}$ in bold is the mean vector, $v$ parameterizes the covariance matrix $\Sigma$, which is chosen to be diagonal. Two parameterizations for the covariance matrix are considered: $\Sigma_{ii} = e^{v_i}$ (log-diagonal) and $\Sigma_{ii} = v_i$ (diagonal). $\psi(x)$ is the sum of $sinc$ functions over all dimensions. Training is up to 4000 iterations, with $\lambda = .9$ and $\beta = .1$ unless they are varied. In Figure 1 (c), $\sigma$ and $\mu$ refer to the std of the first component of the gaussian $\sigma = \sqrt{\Sigma_{11}}$ and $\mu = \boldsymbol{\mu}_1$. More details about the experimental setting are provided in Appendix D.3.

When $q_\theta$ and $q_{\theta+u}$ are the behavioral embedding distributions of two policies $\pi_\theta$ and $\pi_{\theta+u}$, the function $f_u$ allows to transport behavior from a policy $\pi_\theta$ to a behavior as close as possible to $\pi_{\theta+u}$ with the least cost. We thus refer to $f_u$ as the *behavioral transport function*. The function $f_u$ determines how hard it is to change behavior **locally** from policy $\pi_\theta$ in a direction $u$, thus providing a tool to find update directions $u$ with either *maximal* or *minimal* change in behavior.

Probing all directions in a basis $(e_1, ..., e_p)$ of parameters allows us to construct the WIM $G(\theta)$ in Equation (7) which summarizes proximity in behavior along all possible directions $u$ using $u^\top G(\theta)u = \mathbb{E}_{q_\theta}[\|\nabla_x f_u(X)\|^2]$. For an objective $\mathcal{L}(\theta)$, such as the expected total reward of a policy, the Wasserstein natural gradient (WNG) is then defined as the direction $u$ that locally increases $\mathcal{L}(\theta + u)$ the most with the least change in behavior as measured by $f_u$. Formally, the WNG is related to the usual Euclidean gradient $g = \nabla_\theta \mathcal{L}(\theta)$ by

$$g^W = \arg\max_u 2g^\top u - u^\top G(\theta)u. \tag{8}$$

From Equation (8), the WNG can be expressed in closed-form in terms of $G(\theta)$ and $g$ as $g^W = G^{-1}(\theta)g$. Hence, WNG ascent is simply performed using the update equation $\theta_{k+1} = \theta_k + \lambda g_k^W$. We'll see in Section 4 how to estimate WNG efficiently without storing or explicitly inverting the matrix $G$. Next, we discuss the advantages of using WNG over other methods.

## 3.1 WHY USE THE WASSERSTEIN NATURAL GRADIENT?

To illustrate the advantages of the WNG, we consider a simple setting where the objective is of the form $\mathcal{L}(\theta) = \mathbb{E}_{q_\theta}[\psi(x)]$, with $q_\theta$ being a gaussian distribution. The optimal solution in this example is a deterministic point mass located at the global optimum $x^\star$ of the function $\psi(x)$. This situation arises systematically in the context of ES when using a gaussian noise distribution with learnable mean and variance. Moreover, the optimal policy of a Markov Decision Processes (MDP) is necessarily deterministic (Puterman, 2010). Thus, despite its simplicity, this example allows us to obtain closed-form expressions for all methods while capturing a crucial property in many RL problems (deterministic optimal policies) which, as we will see, results in differences in performance.

**Wasserstein natural gradient vs Fisher natural gradient**   While Figure 1 (c) shows that both methods seem to reach the same solution, a closer inspection of the loss, as shown in Figure 1 (d) and (e) for two different parameterizations of $q_\theta$, shows that the FNG is faster at first, then slows down to reach a final error of $10^{-4}$. On the other hand, WNG is slower at first then transitions suddenly to an error of $10^{-8}$. The optimal solution being deterministic, the variance of the gaussian $q_\theta$ needs to shrink to 0. In this case, the KL blows up, while the $W_2$ distance remains finite. As the natural gradient methods are derived from those two divergences (Theorem 2 of Appendix B), they inherit the same behavior. This explains why, unlike the WNG, the FNG doesn't achieve the error of $10^{-8}$. Beyond this example, when the policy $\pi_\theta$ is defined only implicitly using a generative network, as in Tang & Agrawal (2019), the FNG and KL penalty are ill-defined since $\pi_{\theta_k}$ and $\pi_{\theta_{k+1}}$ might have non-overlapping supports. However, the WNG remains well-defined (see Arbel et al. (2020)) and allows for more flexibility in representing policies, such as with behavioral embeddings.

**Wasserstein penalty vs Wasserstein natural gradient**   The Wasserstein penalty Equation (4) encourages *global* proximity between updates $q_{\theta_k}$. For small values of the penalty parameter $\beta$, the method behaves like standard gradient descent (Figure 1 (a)). As $\beta$ increases, the penalty encourages more local updates and thus incorporates more information about the local geometry defined by $q_\theta$. In fact, it recovers the WNG direction (Theorem 2 of Appendix B) albeit with an infinitely small step-size which is detrimental to convergence of the algorithm. To avoid slowing-down, there is an intricate balance between the step-size and penalty $\beta$ that needs to be maintained (Schulman et al., 2017). All of these issues are avoided when directly using the WNG, as shown in Figure 1 (a), which performs the best and tolerates the widest range of step-sizes Figure 1 (f). Moreover, when using the *log-diagonal parameterization* as in Figure 1 (d,a), the WNGD (in red) achieves an error of 1e-8, while $W_2$-penalty achieves a larger error of order 1e-0 for various values of the $\beta$. When using the *diagonal parameterization* instead, as shown in Figure 1 (e), both methods achieve a similar error of 1e-6. This discrepancy in performance highlights the robustness of WNG to parameterization of the model.

**Combining WNG and a Wasserstein penalty.**   The global proximity encouraged by a $W_2$ penalty can be useful on its own, for instance, to explicitly guarantee policy improvement as in (Pacchiano et al., 2019, Theorem 5.1). However, this requires estimating the $W_2$ at every iteration, which can be costly. Using WNG instead of the usual gradient can yield more efficient updates, thus reducing the number of time $W_2$ needs to be estimated. The speed-up can be understood as performing second-order optimization on the $W_2$ penalty since the WNG arises precisely from a second-order expansion of the $W_2$ distance, as shown in Section 3 (See also Example 2 in Arbel et al. (2020)).

## 4   POLICY OPTIMIZATION USING BEHAVIORAL GEOMETRY

We now present practical algorithms to exploit the behavioral geometry induced by the embeddings $\Phi$. We begin by describing how to efficiently estimate the WNG.

**Efficient estimation of the WNG**   can be performed using kernel methods, as shown in Arbel et al. (2020) in the case where the re-parametrization trick is applicable. This is the case, if for instance, the behavioral variable is the concatenation of actions $X = [a_0, ..., a_T]$ and if actions are sampled from a gaussian with mean and variance parameterized by a neural network, as is often done in practice for real-valued actions. Then $X$ can be expressed as $X = B_\theta(Z)$ where $B_\theta$ is a known function and $Z$ is an input sample consisting in the concatenation of states $[s_0, ..., s_T]$ and the gaussian noise used to generate the actions. However, the proposed algorithm is not readily applicable if for instance the behavioral variable $X$ is a function of the reward.

We now introduce a procedure that extends the previous method to more general cases, including those where only the score $\nabla_\theta \log q_\theta$ is available without an explicit re-parametrization trick. The core idea is to approximate the functions $f_{e_j}$ defining $G(\theta_k)$ in Equation (7) using a linear combinations of user-specified basis functions $(h_1(x), ..., h_M(x))$:

$$\hat{f}_{e_j}(x) = \sum_{m=1}^{M} \alpha_m^j h_m(x), \qquad (9)$$

The number $M$ controls the computational cost of the estimation and is typically chosen on the order of $M = 10$. The basis can be chosen to be *data-dependent* using kernel methods. More precisely,

---

Algorithm 1: Wasserstein Natural Policy Gradient

1: **Input** Initial policy $\pi_{\theta_0}$
2: **for** iteration $k = 1, 2, ...$ **do**
3:    Obtain $N$ rollouts $\{\tau\}_{n=1}^N$ of length $T$ using policy $\pi_{\theta_k}$
4:    Compute loss $\mathcal{L}(\theta_k)$ in a forward pass
5:    Compute gradient $\hat{g}_k$ in the backward pass on $\mathcal{L}(\theta_k)$
6:    Compute Behavioral embeddings $\{X_n = \Phi(\tau^n)\}_{n=1}^N$
7:    Compute WNG $\hat{g}_k^W$ using Algorithm 3 with samples $\{X_n\}_{n=1}^N$ and gradient estimate $\hat{g}_k$.
8:    Update policy using: $\theta_{k+1} = \theta_k + \lambda \hat{g}_k^W$.
9: **end for**

---

we use the same approach as in Arbel et al. (2020), where we first subsample $M$ data-points $Y_m$ from a batch of $N$ variables $X_n$ and $M$ indices $i_m$ from $\{1, ..., d\}$ where $d$ is the dimension of $X_n$. Then, each basis can of the form $h_m(x) = \partial_{i_m} K(Y_m, x)$ where $K$ is a positive semi-definite kernel, such as the gaussian kernel $K(x, y) = \exp(-\frac{\|x-y\|^2}{\sigma^2})$. This choice of basis allows us to provide guarantees for functions $f_j$ in terms of the batch size $N$ and the number of basis points $M$ (Arbel et al., 2020, Theorem 7). Plugging-in each $\hat{f}_j$ in the transport cost problem Equation (5) yields a quadratic problem of dimension $M$ in the coefficients $\alpha^j$:

$$\text{maximize}_{\alpha^j}\ 2J_{\cdot,j}\alpha^j - (\alpha^j)^\top L \alpha^j$$

where $L$ is a square matrix of size $M \times M$ independent of the index $j$ and $J$ is a Jacobian matrix of shape $M \times p$ with rows given by $J_{m,\cdot} = \nabla_\theta \mathbb{E}_{q_{\theta_k}}[h_m(X)]$. There are two expressions for $J$, depending on the applicability of the re-parametrization trick or the availability of the score

$$J_{m,\cdot} = \hat{\mathbb{E}}_{q_\theta}[\nabla_x h_m(X)\nabla_\theta B_\theta(Z)] \qquad \text{or} \qquad J_{m,\cdot} = \hat{\mathbb{E}}_{q_\theta}[\nabla_\theta \log q_\theta(X) h_m(X)] \qquad (10)$$

Computing $J$ can be done efficiently for moderate size $M$ by first computing a surrogate vector of $V$ of size $M$ whose Jacobian recovers $J$ using automatic differentiation software:

$$V_m = \hat{\mathbb{E}}_{q_\theta}[h_m(X_n)], \qquad \text{or} \qquad V_m = \hat{\mathbb{E}}_{q_\theta}[\log q_\theta(X_n) h_m(X_n)]. \qquad (11)$$

The optimal coefficients $\alpha^j$ are then simply expressed as $\alpha = L^\dagger J$. Plugging-in the optimal functions in the expression of the Wasserstein Information Matrix (Equation (7)), yields a low rank approximation of $G$ of the form $\hat{G} = J^\top L^\dagger J$. By adding a small diagonal perturbation matrix $\epsilon I$, it is possible efficiently compute $(\hat{G} + \epsilon I)^{-1}\hat{g}$ using a generalized Woodbury matrix identity which yields an estimator for the Wasserstein Natural gradient

$$\hat{g}^W = \frac{1}{\epsilon}\left(\hat{g} - J^\top \left(JJ^\top + \epsilon L\right)^\dagger J\hat{g}\right). \qquad (12)$$

The pseudo-inverse is only computed for a matrix of size $M$. Using the Jacobian-vector product, Equation (12) can be computed without storing large matrices $G$ as shown in Algorithm 3.

**Wasserstein Natural Policy Gradient (WNPG).** It is possible to incorporate local information about the behavior of a policy in standard algorithms for policy gradient as summarized in Algorithm 1. In its simplest form, one first needs to compute the gradient $\hat{g}_k$ of the objective $\mathcal{L}(\theta_k)$ using, for instance, the REINFORCE estimator computed using $N$ trajectories $\tau_n$. The trajectories are then used to compute the BEMs which are fed as input, along with the gradient $\hat{g}_k$ to get an estimate of the WNG $g_k^W$. Finally, the policy can be updated in the direction of $g_k^W$. Algorithm 1 can also be used in combination with an explicit $W_2$ penalty to control non-local changes in behavior of the policy thus ensuring a policy improvement property as in (Pacchiano et al., 2019, Theorem 5.1). In that case, WNG enhances convergence by acting as a second-order optimizer, as discussed in Section 3.1. The standard gradient $\hat{g}_k$ in Algorithm 1 is then simply replaced by the one computed in (Pacchiano et al., 2019, Algorithm 3). In Section 5, we show that this combination, which we call behavior-guided WNPG (BG-WNPG), leads to the best overall performance.

**Wasserstein Natural Evolution Strategies (WNES).** ES treats the total reward observed on a trajectory under policy $\pi_\theta$ as a black-box function $\mathcal{L}(\theta)$ (Salimans et al., 2017; Mania et al., 2018;

---

Algorithm 2: Wasserstein Natural Evolution Strategies

---

1: **Input** Initial policy $\pi_{\theta_0}$, $\alpha > 0$, $\delta \leq 1$
2: **for** iteration $k = 1, 2, ...$ **do**
3:     Sample $\epsilon_1, \ldots, \epsilon_n \sim \mathcal{N}(0, I)$.
4:     Perform rollouts $\{\tau_n\}_{n-1}^N$ of length $T$ using the perturbed parameters $\{\widetilde{\theta}^n = \theta_k + \sigma\epsilon_n\}_{n=1}^N$
       and compute behavioral embeddings $\{X_n = \Phi(\tau^n)\}_{n=1}^N$
5:     Compute gradient estimate of $\mathcal{L}(\widetilde{\theta}^n)$ using Equation (13) and trajectories $\{\tau^n\}_{n=1}^N$.
6:     Compute Jacobian matrix $J$ appearing in Algorithm 3 using Equation (14).
7:     Compute WNG $\hat{g}_k^W$ using Algorithm 3, with samples $\{X_n\}_{i=1}^N$ and computed $\hat{g}_k$ and $J$.
8:     Update policy using Equation (15).
9: **end for**

---

Choromanski et al., 2020). Evaluating it under $N$ policies whose parameters $\widetilde{\theta}^n$ are gaussian perturbations centered around $\theta_k$ and with variance $\sigma$ can give an estimate of the gradient of $\mathcal{L}(\theta_k)$:

$$\hat{g}_k = \frac{1}{N\sigma} \sum_{n=1}^N \left( \mathcal{L}(\widetilde{\theta}^n) - \mathcal{L}(\theta_k) \right) (\widetilde{\theta}^n - \theta_k). \tag{13}$$

Instead of directly updating the policy using Equation (13), it is possible to encourage either proximity or diversity in behavior using the embeddings $X_n = \Phi(\tau_n)$ of the trajectories $\tau_n$ generated for each perturbed policy $\pi_{\widetilde{\theta}_n}$. Those embeddings can be used as input to Algorithm 3 (see appendix), along with Equation (13) to estimate the $\hat{g}_k^W$, which captures similarity in behavior. The algorithm remains unchanged except for the estimation of the Jacobian $J$ of Equation (10) which becomes

$$J_{m,.} = \frac{1}{N\sigma} \sum_{n=1}^N h_m(X_n)(\widetilde{\theta}^n - \theta_k). \tag{14}$$

The policy parameter can then be updated using an interpolation between $\hat{g}_k$ and the WNG $\hat{g}_k^W$, i.e.,

$$\Delta\theta_k \propto (1 - \delta)\hat{g}_k + \delta\hat{g}_k^W \tag{15}$$

with $\delta \leq 1$ that can also be negative. Positive values for $\delta$ encourage proximity in behavior, the limit case being $\delta = 1$ where a full WNG step is taken. Negative values encourage repulsion and therefore need to compensated by $\hat{g}_k$ to ensure overall policy improvement. Algorithm 2 summarizes the whole procedure, which can be easily adapted from existing ES implementations by calling a variant of Algorithm 3. In particular, it can also be used along with an explicit $W_2$ penalty, in which case the proposed algorithm in Pacchiano et al. (2019) is used to estimate the standard gradient $\hat{g}_k$ of the penalized loss. Then the policy is updated using Equation (15) instead of $\hat{g}_k$. We refer to this approach as behavior-guided WNES (BG-WNES).

## 5 EXPERIMENTS

We now test the performance of our estimators for both policy gradients (PG) and evolution strategies (ES) against their associated baseline methods. We show that in addition to an improved computational efficiency, our approach can effectively utilize the geometry induced by a Wasserstein penalty to improve performance, particularly when the optimization problem is ill-conditioned. Further experimental details can be found in the appendix, and our code is available online[1].

**Policy Gradients.** We first apply WNPG and BG-WNPG to challenging tasks from OpenAI Gym (Brockman et al., 2016) and Roboschool (RS). We compare performance against behavior-guided policy gradients (BGPG), (Pacchiano et al., 2019), PPO with clipped surrogate objective (Schulman et al., 2017) (PPO (Clip)), and PG with no trust region (None). From Figure 2, we can see that BGPG outperforms the corresponding KL-based method (PPO) and vanilla PG, as also demonstrated in the work of Pacchiano et al. (2019). Our method (WNPG) matches or exceeds final performance of BGPG on all tasks. Moreover, combining both (BG-WNPG) produces the largest gains on all

---

[1]https://github.com/tedmoskovitz/WNPG

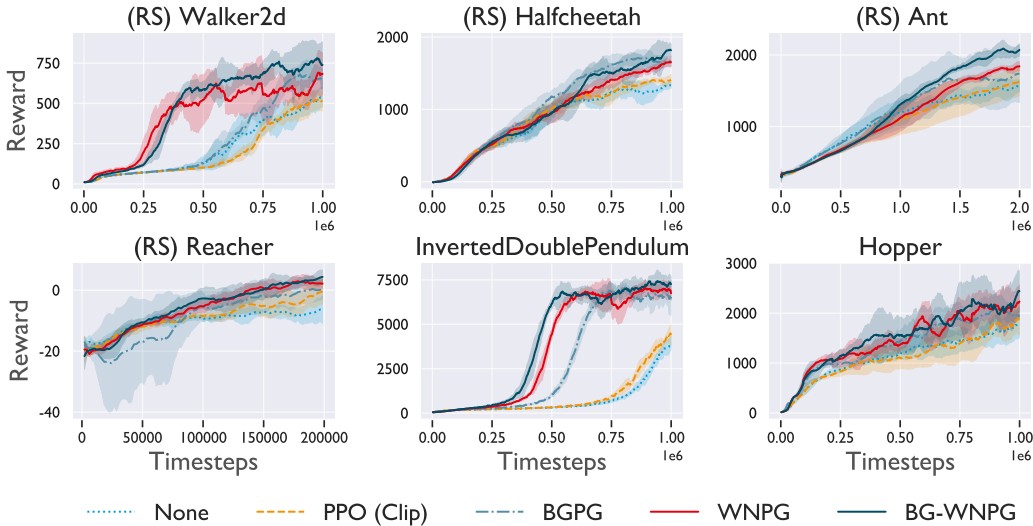

Figure 2: **WNG-based algorithms provide large gains on tasks where initial progress is difficult.** The performance mean $\pm$ standard deviation is plotted versus time steps for 5 random seeds on each task.

environments. Final mean rewards are reported in Table 1. It is also important to note that WNG-based methods appear to offer the biggest advantage on tasks where initial progress is difficult. To investigate this further, we computed the hessian matrix at the end of training for each task and measured the ratios of its largest eigenvalue to each successive eigenvalue (Figure 3). Larger ratios indicate ill-conditioning, and it is significant that WNG methods produce the greatest improvement on the environments with the poorest conditioning. This is consistent with the findings in Arbel et al. (2020) that showed WNG to perform most favorably compared to other methods when the optimization problem is ill-conditioned, and implies a useful heuristic for gauging when WNG-based methods are most useful for a given problem.

**Evolution Strategies** To test our estimator for WNES, as well as BG-WNES, we applied our approach to the environment introduced by Pacchiano et al. (2019), designed to test the ability of behavior-guided learning to succeed despite deceptive rewards. During the task, the agent receives a penalty proportional to its distance from a goal, but a wall is placed directly in the agent's path (Figure 7). This barrier induces a local maximum in the objective—a naïve agent will simply walk directly towards the goal and get stuck at the barrier. The idea is that the behavioral repulsion fostered by applying a positive coefficient to the Wasserstein penalty ($\beta > 0$) will encourage the agent to

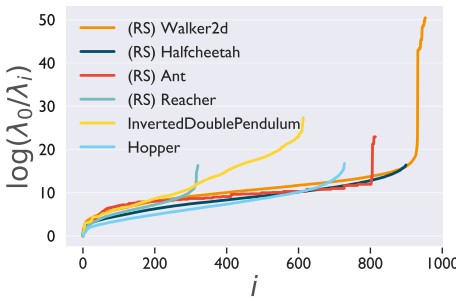

Figure 3: Condition numbers for different tasks.

seek novel policies, helping it to eventually circumvent the wall. As in Pacchiano et al. (2019), we test two agents, a simple point and a quadruped. We then compare our method with vanilla ES as described by Salimans et al. (2017), ES with gradient norm clipping, BGES (Pacchiano et al., 2019), and NSR-ES (Conti et al., 2018a). In Figure 4, we can see that WNES and BG-WNES improve over the baselines for both agents. To test that the improvement shown by BG-WNES wasn't simply a case of additional "repulsion" supplied by the WNG to BGES, we also tested BGES with an increased $\beta = 0.75$, compared to the default of $0.5$. This resulted in a decrease in performance, attesting to the unique benefit provided by the WNES estimator.

**Computational Efficiency** We define the *computational efficiency* of an algorithm as the rate with which it accumulates reward relative to its runtime. To test the computational efficiency of our

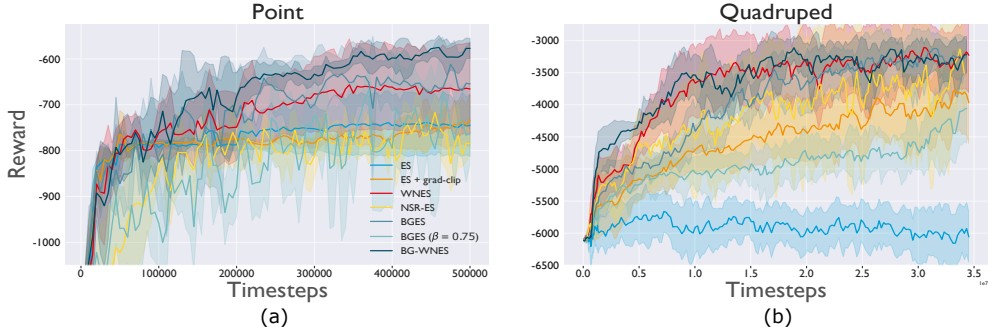

Figure 4: **WNES methods more reliably overcome local maxima.** Results obtained on the point (a) and quadruped (b) tasks. The mean ± standard deviation is plotted across 5 random seeds.

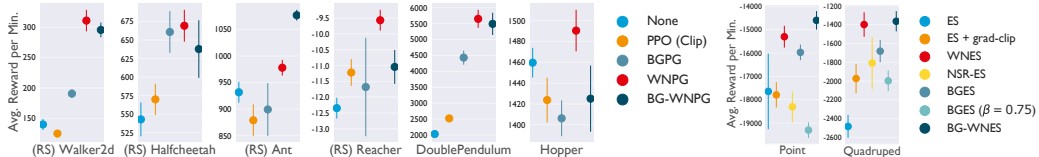

Figure 5: **WNG methods improve computational efficiency.** Average reward per minute is plotted for both gradient tasks (left) and ES tasks (right) for the runs depicted above.

approach, we plotted the total reward divided by wall clock time obtained by each agent for each task (Fig. 5). Methods using a WNG estimator were the most efficient on each task for both PG and ES agents. On several environments used for the policy gradient tasks, the added cost of BG-WNPG reduced its efficiency, despite having the highest absolute performance.

## 6  CONCLUSION

Explicit regularization using divergence measures between policy representations has been a common theme in recent work on policy optimization for RL. While prior works have previously focused on the KL divergence, Pacchiano et al. (2019) showed that a Wasserstein regularizer over behavioral distributions provides a powerful alternative framework. Both approaches implicitly define a form of natural gradient, depending on which divergence measure is chosen. Through the introduction of WNPG and WNES, we demonstrate that directly estimating the natural gradient of the un-regularized objective can deliver greater performance at lower computational cost. These algorithms represent novel extensions of previous work on the WNG to problems where the reparameterization trick is not available, as well as to black-box methods like ES. Moreover, using the WNG in conjunction with a WD penalty allows the WNG to take advantage of the local geometry induced by the regularization, further improving performance. We also provide a novel comparison between the WNG and FNG, showing that the former has significant advantages on certain problems. We believe this framework opens up a number of avenues for future work. Developing a principled way to identify useful behavioral embeddings for a given RL task would allow to get the highest benefit form WNPG and WNES. From a theoretical perspective, it would be useful to characterize the convergence boost granted by the combination of explicit regularization and the corresponding natural gradient approach.

**Acknowledgments** The authors would like to thank Jack Parker-Holder for sharing his code for BGPG and BGES, as well as colleagues at Gatsby for useful discussions.

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

| Environment | BG-WNPG (ours) | WNPG (ours) | BGPG | PPO (Clip) | None |
|---|---|---|---|---|---|
| (RS) Walker2d | **739.51±81.10** | 683.38⋆ | 652.53 | 516.47 | 529.60 |
| (RS) Halfcheetah | **1817.68 ± 79.64** | 1655.04 | 1668.75* | 1412.57 | 1334.27 |
| (RS) Ant | **2072.56 ± 85.24** | 1844.14* | 1735.98 | 1627.01 | 1566.63 |
| (RS) Reacher | **4.37 ± 2.23** | 2.13* | 0.46 | -0.37 | -6.46 |
| DoublePendulum | **7254.98 ± 419.36** | 6740.96* | 6526.48 | 4401.24 | 3964.51 |
| Hopper | **2439.56 ± 394.39** | 2235.23* | 1934.43 | 1890.12 | 1816.58 |

Table 1: Final average return over 5 trials for the PG experiments depicted in Figure 2. ± values denote one standard deviation across trials. The value for the best-performing method is listed in **bold**, while a * denotes the second best-performing method. BG-WNPG reaches the highest performance on all tasks. WNPG beats the best-performing baseline (BGPG) on all tasks except HalfCheetah, where the difference is small.

## A  BACKGROUND

### A.1  POLICY OPTIMIZATION

An agent interacting with an environment form a system that can be described by a *state* variable $s$ belonging to a *state space* $\mathcal{S}$. In the Markov Decision Process (MDP) setting, the agent can interact with the environment by taking an action $a$ from a set of possible actions $\mathcal{A}$ given the current state $s$ of the system. As a consequence, the system moves to a new state $s'$ according to a probability transition function $P(s'|a, s)$ which describes the probability of moving to state $s'$ given the previous state $s$ and action $a$. The agent also receives a partial reward $r$ which can be expressed as a possibly randomized function of the new state $s'$, $r = r(s')$. The agent has access to a set of possible *policies* $\pi_\theta(a|s)$ parametrized by $\theta \in \mathbb{R}^p$ and that generates an action $a$ given a current state $s$. Thus, each policy can be seen as a probability distribution conditioned a state $s$. Using the same policy induces a whole trajectory of state-action-rewards $\tau = (s_t, a_t, r_t)_{t \geq 0}$ which can be viewed as a sample from a trajectory distribution $\mathbb{P}_\theta$ defined over the space of possible trajectories $\tau$. Hence, for a given random trajectory $\tau$ induced by a policy $\pi_\theta$, the agent receives a total discounted reward $R(\tau) := \sum_{t=1}^{\infty} \gamma^{t-1} r(s_t)$ with discount factor $0 < \gamma < 1$. This allows to define the *value* function as the expected total reward conditioned on a particular initial state $s$:

$$V_\theta(s_t) = \mathbb{E}_{\mathbb{P}_\theta|s_t} \left[ \sum_{l=1}^{\infty} \gamma^{l-1} r(s_{l+t}) \right]. \tag{16}$$

When the gradient of the *score function* $\nabla \log \pi_\theta(a|s)$ is available, the policy gradient theorem allows us to express the gradient of $\mathcal{R}(\theta)$:

$$\nabla_\theta \mathcal{R}(\theta) = \mathbb{E}_{\mathbb{P}_\theta} \left[ \sum_{t=0}^{\infty} \gamma^t \nabla \log \pi_\theta(a_t|s_t) A_\theta(s_t, a_t) \right] \tag{17}$$

where the expectation is taken over trajectories $\tau$ under $\mathbb{P}_\theta$ and $A_\theta(s, a)$ represents the *advantage* function which can be expressed in terms of the value function $V_\theta(s)$ in terms of

$$A_\theta(s_t, a_t) = \mathbb{E}_{s_{t+1}|s_t, a_t} \left[ r(s_{t+1}) + \gamma V_\theta(s_{t+1}) \right] - V_\theta(s_t).$$

The agent seeks an optimal policy $\pi_{\theta^\star}$ that maximizes the expected total reward under the trajectory distribution: $\mathcal{R}(\theta) = \mathbb{E}_{\mathbb{P}_\theta}[R(\tau)]$.

## B  WASSERSTEIN NATURAL GRADIENT

**Connection to the Fisher natural gradient and proximal methods.**  Both WNG and FNG are obtained from a proximity measure between probability distributions:

**Proposition 2** *Let $D(\theta, \theta')$ be either the KL-divergence $KL(\pi_\theta, \pi_{\theta'})$ or the Wasserstein-2 distance between the behavioral distributions $W_2(q_\theta, q_{\theta'})$ and let $g^D$ be either the FNG $g^F$ or WNG $g^W$,*

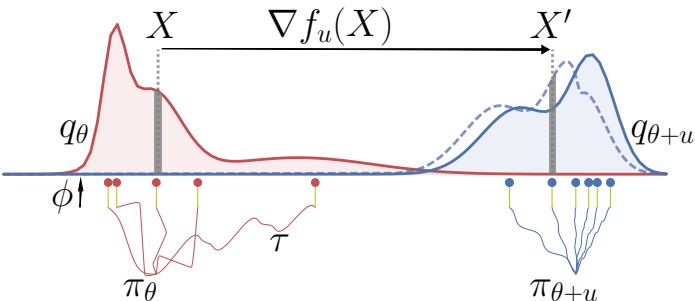

Figure 6: A visualization of the behavioral transport function.

*then*

$$g_k^D = \lim_{\beta \to +\infty} \arg\max_u \beta \left( \mathcal{L}(\theta_k + \beta^{-1}u) - \mathcal{L}(\theta_k) - \frac{\beta}{2} D\left(\theta_k, \theta_k + \beta^{-1}u\right) \right) \tag{18}$$

Equation (18) simply states that the both WNG and FNG arise as limit cases of penalized objectives provided the strength of the penalty $\beta$ diverges to infinity and the step-size is shrank proportionally to $\beta^{-1}$. An additional global rescaling by $\beta$ of the total objective prevents it from collapsing to 0. Intuitively, performing a Taylor expansion of Equation (18) recovers an equation similar to Equation (8). Equation (18) shows that using a penalty that encourages **global** proximity between successive policies, it is possible to recover the **local** geometry of policies (captured by the local ) by increasing the strength of the penalty using appropriate re-scaling. This also informally shows why both natural gradients are said to be invariant to re-parametrization (Arbel et al., 2020, Proposition 1), since both KL and $W_2$ remains unchanged if $q_\theta$ is parameterized in a different way.

## C   ALGORITHM FOR ESTIMATING WNG

---

Algorithm 3:  Efficient Wasserstein Natural Gradient

---

1: **Input** mini-batch of samples $\{X_n\}_{n=1}^N$ distributed according to $q_\theta$, gradient direction $\hat{g}$, basis functions $h_1, ..., h_M$, regularization parameter $\epsilon$.
2: **Output** Wasserstein Natural gradient $\hat{g}^W$
3: Compute a matrix $C$ of shape $M \times Nd$ using $C_{m,(n,i)} \leftarrow \partial_i h_m(X_n)$.
4: Compute similarity matrix $L \leftarrow \frac{1}{N} CC^T$.
5: Compute surrogate vector $V$ using Equation (11).
6: **for** iteration$= 1, 2, ...M$ **do**
7:    Use automatic differentiation on $V_m$ to compute Jacobian matrix $J$ in Equation (10).
8: **end for**
9: Compute a matrix $D$ of shape $M \times M$ using $D \leftarrow JJ^\top + \epsilon L$.
10: Compute a vector $b$ of size $M$ using $b \leftarrow J\hat{g}$.
11: Solve linear system of size $M : b \leftarrow \texttt{solve}\,(D, b)$
12: Return $\hat{g}^W \leftarrow \frac{1}{\epsilon}(\hat{g} - J^\top b)$

---

## D   ADDITIONAL EXPERIMENTAL DETAILS

### D.1   POLICY GRADIENT TASKS

We conserve all baseline and shared hyperparameters used by Pacchiano et al. (2019). More precisely, for each task we ran a hyperparameter sweep over learning rates in the set $\{$1e-5, 5e-5, 1e-4, 3e-4$\}$, and used the concatenation-of-actions behavioral embedding $\Phi(\tau) = [a_0, a_1, \dots, a_T]$ with the base network implementation the same as Dhariwal et al. (2017).

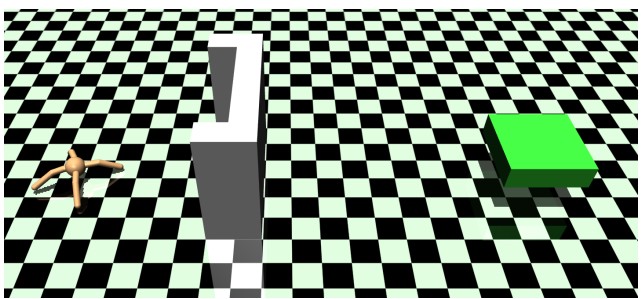

Figure 7: A visualization of the quadruped task. The agent receives receives more reward the closer it is to the goal (green). A naïve agent will get stuck in the local maximum at the wall if it attempts to move directly to the goal.

The WNG hyperparameters were also left the same as in Arbel et al. (2020). Specifically, the number of basis points was set as $M = 5$, the reduction factor was bounded in the range $[0.25, 0.75]$, and $\epsilon \in [1\text{e-}10, 1\text{e}5]$.

## D.2 EVOLUTION STRATEGIES TASKS

As with the policy gradient tasks, we conserved all baseline and shared hyperparameters used by Pacchiano et al. (2019). Specifically, for the point task, we set the learning rate to be $\eta = 0.1$, the standard deviation of the noise to be $\sigma = 0.01$, the rollout length $H$ was 50 time steps, and the behavioral embedding function to be the last state $\Phi(\tau) = s_H$. For the quadruped task we set $\eta = 0.02$, $\sigma = 0.02$, $H = 400$, and $\Phi(\tau) = \sum_{t=0}^{H} r_t \left( \sum_{i=0}^{t} e_i \right)$ (reward-to-go encoding; see Pacchiano et al. (2019) for more details). Both tasks used 1000-dimensional random features and embeddings from the $n = 2$ previous policies to compute the WD.

For WNG, the same hyperparameters were used as in the policy gradient tasks.

## D.3 EXPERIMENTAL SETTING OF FIGURE 1

**The Objective**  We consider a function $\psi(x)$ is the sum of $sinc$ functions over all dimensions of $x \in \mathbb{R}^{100}$

$$\psi(x) = \sum_{i=1}^{100} \frac{sin(x_i)}{x_i} - 1 \tag{19}$$

Such function is highly non-convex and admits multiple bad local minima with the global minimum of $\psi(x)$ reached for $x^\star = 0$. However, we do not make use of this information during optimization. To alleviate the non-convexity of this loss, we consider a gaussian relaxation objective $\mathcal{L}(\theta)$ obtained by taking the expectation of $\psi(x)$ over the 100 dimensional vector $x$ w.r.t. to a gaussian $q_\theta$ with parameter vector $\theta$. Thus the objective function to be optimized is a function of $\theta$:

$$\mathcal{L}(\theta) = \mathbb{E}_{q_\theta}[\psi(x)] \tag{20}$$

The parameter vector $\theta$ is of the form $\theta = (\mu, v)$, where $\mu$ is the mean of the gaussian $q_\theta$ and $v$ is a vector in $\mathbb{R}^{100}$ parameterizing the covariance matrix $\Sigma$ of the gaussian $q_\theta$. We will later consider two parameterizations for the covariance matrix.

The minimal value of $\mathcal{L}(\theta)$ is reached when the gaussian $q_\theta$ is degenerate with $\Sigma = 0$ and mean $\mu = x^\star = 0$. Hence, the mean parameter of the global minimum of $\mathcal{L}(\theta)$ recover the global optimum of $\psi$.

**Parameterization of the gaussian**  We choose the covariance matrix of the gaussian to be diagonal and consider two parameterizations for the covariance matrix $\Sigma$: *diagonal* and *log-diagonal*. For the *diagonal* parameterization the Covariance $\Sigma_{ii} = v_i$ and for the *log-diagonal* we set $\Sigma_{ii} = \exp(2v_i)$.

**Optimization methods**    We consider different optimization methods using the same objective $\mathcal{L}(\theta)$. For the penalty methods, we use the closed form expressions for the both the Wasserstein distance and KL which are available explicitly in the case of gaussians.

For the Natural gradient methods (WNG) and (FNG), we use the closed form expressions which are also available in the gaussian case. We denote them as $\nabla^W \mathcal{L}(\theta)$ for (WNG) and $\nabla^F \mathcal{L}(\theta)$ for FNG and express them in terms of the euclidean/standard gradient $\nabla \mathcal{L}(\theta)$:

- **Diagonal parameterization:**
  - WNG:
  $$\nabla_v^W \mathcal{L}(\theta) = 4 * \Sigma \nabla_v \mathcal{L}(\theta), \qquad \nabla_\mu^W \mathcal{L}(\theta) = \nabla_\mu \mathcal{L}(\theta) \tag{21}$$
  - FNG:
  $$\nabla_v^F \mathcal{L}(\theta) = 2\Sigma^2 \nabla_\Sigma \mathcal{L}(\theta), \qquad \nabla_\mu^F \mathcal{L}(\theta) = \Sigma \nabla_\mu \mathcal{L}(\theta) \tag{22}$$
- **Log-diagonal parameterization:**
  - WNG:
  $$\nabla_v^W \mathcal{L}(\theta) = \Sigma^{-1} \nabla_v \mathcal{L}(\theta), \qquad \nabla_\mu^W \mathcal{L}(\theta) = \nabla_\mu \mathcal{L}(\theta) \tag{23}$$
  - FNG:
  $$\nabla_v^F \mathcal{L}(\theta) = .5 * \nabla_v \mathcal{L}(\theta), \qquad \nabla_\mu^F \mathcal{L}(\theta) = \Sigma \nabla_\mu \mathcal{L}(\theta) \tag{24}$$

**Training details**    Training is up to 4000 gradient iterations, with $\lambda = .9$ and $\beta = .1$ unless they are varied.

