# OpenReview forum: "Efficient Wasserstein Natural Gradients for Reinforcement Learning"
_ICLR.cc/2021/Conference — ICLR 2021 Poster_

### Official Review · AnonReviewer1 · 2020-10-27
**An encouraging empirical result but low technical novelty and seemly insufficient experiments to make a reliable conclusion given this work is mostly empirical**

**Rating:** 6
**Confidence:** 4

**Review:**

### Summary
This paper proposes to use natural gradient instead of standard gradient to optimize a regularized objective with the regularization being the Wasserstein distance between the so-called behaviour distributions for the previous policy and new policy. It then combines this Wasserstein gradient descent with Policy Gradient and Evolutionary Strategies.  Experiments conducted in OpenAI and Roboschool show some promising results for this combination.

### Strong points:
-	Clarity: The paper is well structured
-	Empirical significance: The empirical results seem promising where it shows that Wasserstein gradient descent could be a good choice to constraint the behaviour changes efficiently without sacrificing too much computational cost. I personally like that the authors used Roboschool (RS) as a benchmark for continuous control tasks as it is, as opposed to Mujuco, a free simulator software (thus it is more accessible and reproducible for everyone).

### Weak points:
-	Novelty: the work however has low technical novelty where it combines several known results into a new framework. In particular, the idea of constraining behaviour policy changes and the efficient way to estimate Wasserstein gradient descent are all known and off-the-shelf. Adopting Wasserstein gradient descent to RL constraint update seems straightforward that does not require any significant technical challenge. The interpretation of the framework also seems straightforward, e.g., it is of course that updating along the Wasserstein natural gradient would incorporate the local geometry of parameterization and help overcome some ill-conditioning issues where KL has.
-	Empirical significance: The empirical results though promising are not strong given that this is mostly an empirical work. In particular, the present work presents the experiments for PG case in only 4 environments which I think insufficient to make a reliable conclusion about its empirical significance.

###  Questions for the authors
-	In Section 2: “Reusing trajectories can reduce the computational cost but drastically increases the variance of the gradient estimator”. Could the authors elaborate on why reusing trajectories drastically increases the variance of the gradient estimator?
-	Fig. 1 (c): Have all algorithms been initialized at the same initial point? Also, according to Fig.1 (c) that it seems that FNGD has a ‘right’ convergence when it converges to the point where \sigma=0 and \mu = midpoint, why in the last paragraph of page 4, the authors conclude that “FNG remains far away from optimum”? What do I miss here?
-	What is the difference between W2-penality and WNGD in Figure 1?
-	On page 5, “The Wasserstein penalty Equation (4) encourages global proximity between updates qθk”. What does globality here refer to while Eq (6) holds only locally?

### Minor comments
-	The second term of Eq. (5): Shouldn’t it be f_u instead of f there?
-	Eq. (8): \argmax_{u}
-	The second last sentence at the end of page 4: cite -> \cite{sth}
-	It seems that \mu and \sigma in Fig. 1 (c) have not defined explicitly anywhere. In the capture of Fig.1., it writes \theta = (mu,v), so there is a chance of inconsistent notations here?
-	On page 5, “To avoid slowing-down, there is an intricate balance between the step-size and penalty β that needs to be maintained Schulman et al.
(2017)”: \cite -> \citep

###  My initial recommendation

Given the weak and strong points above, I vote for rejecting for this current form.

### My finial recommendation

After the discussion and revision, the authors have presented more convincingly and more clearly the empirical significance and applicability of their method. I highly recommend the authors to highlight the lastest discussion in the final paper, especially the ill-conditioned argument, as it is highly relevant to the practitioners. I think this paper can be interesting for a moderate number of readers, especially the use of the open-sourced Roboschool could also increase its reproduciability. I agree to increas my score to 6.

---

> ### Author Response · Authors · 2020-11-17
> **Authors' Response Pt. 1**
>
> Thank you for your detailed comments and for the insightful feedback. We are happy to hear that you found the approach promising and the paper to be clear and well structured.
> We hope the following response addresses your concerns.
>
>
> 1. Novelty: “The work however has low technical novelty where it combines several known results into a new framework”. We respectfully disagree.
> This work indeed relies on three important and very recent ideas: “ Behavioral embeddings for RL” from Pacchiano  et al. (2020), the “Wasserstein Natural Gradient (WNG)” (Li 2019) and the scalable estimator of the WNG from (Arbel 2020). Our main contribution is to build on those works to propose general and flexible methods for Policy gradient and Evolution Strategies with broad applicability in RL. Note also that WNG is a rather novel method that remains under-explored in machine learning literature. Our work shows that this new tool can be useful for RL and black-box methods like ES.  Moreover, as a second contribution, we also extended previous work for estimating WNG in (Arbel 2020) to cases where only the score function is available.  We thus believe the methods we proposed are novel and of general interest to the RL community. Using the Wasserstein Natural Gradient directly preconditions the ‘usual’ gradient of the objective to incorporate local geometry and help overcome ill-conditioning. While this might seem straightforward now, we are not aware of any prior work that exploited this fact in RL (Please see also answer the point (4) below about Globality vs Locality).  As a comparison, we do not believe that TRPO was straightforward before it  was introduced and showed to yield improvement in RL.
>
> 2. Technical challenge: “Adopting Wasserstein gradient descent to RL constraint update seems straightforward that does not require any significant technical challenge.”
> Thank you, we indeed tried our best to make the algorithms seem easy to use off-the-shelf. By doing so, we hope it will reach a wider community. This keeping in mind that the proposed methods rely on several advanced mathematical concepts such as optimal transport, differential geometry, kernel methods. We believe we  provided an accessible and simple presentation of those concepts.
>
>
> 3. Empirical Significance: We are very glad to hear that you see our initial results as promising. We also thank you for encouraging us to strengthen our empirical results. We  have now added new results for the policy gradient methods on two new Roboschool environments (Ant and Reacher) to our revised submission. WNG-based methods outperform baselines on both. We also performed eigenspectra analysis further confirming consistency with the theoretical prediction that WNG-based approaches deliver larger benefits on ill-conditioned problems. We believe that these additional experiments provide further support for the reliability and empirical significance of our approach.

---

> > ### Author Response · Authors · 2020-11-17
> > **Authors' Response Pt. 2**
> >
> > **Answers to Questions:**
> >
> > 1.- “Reusing trajectories”: Thank you,  we have now clarified this statement in the revised version of the document by saying:
> > “Performing multiple policy updates while using trajectories from an older policy  $\pi_{\theta_{old}}$ means that current policy $\pi{\theta}$ can drift away from the older policy. On the other hand, the objective is  obtained as an expectation under $\pi_{\theta}$ for which fresh trajectories are not available. Instead, the objective is estimated using importance sampling (by reweighting the old trajectories according to importance weights $ \frac{\pi_{\theta}}{\pi_{\theta_{old}}} $ ).  When $\pi{\theta}$ is too far from  $\pi_{\theta_{old}}$ , the importance weight has a large variance.”
> >
> > We also referred to Section 2.1 of Schulman et al. (2017), which also discussed this
> >  phenomenon.
> >
> >
> > 2. Figure 1c:  We apologize for this confusion. We clarified this in the revised version in paragraph 2 of section 3.1 by saying:
> > “While  Figure1 (c) shows that both methods seem to reach the same solution, a closer inspection of the loss,
> > 	as shown in Figure1 (d) and (e) for two different parameterizations of $q_{\theta}$, indicates that FNG is faster at first then slows down to reach a final error of $10^{-4}$. On the other hand, WNG is slower at first then improves quickly to an error of $10^{-8}$.
> > ”
> > Thus, in this case, if one would like to optimize the objective to a higher precision, we would recommend WNG.
> >
> > 3. We added the following discussion to the 3rd paragraph of section 3.1 to clarify the  difference between W2-penalty and WNGD in figure 1:
> > “In figure 1(a), the WNGD (in red) achieves an error of 1e-8, while W2-penalty achieves a larger error of order 1e-0 for various values of the $\beta$. This is when using the **log-diagonal parameterization**. When using the **diagonal parameterization** instead as shown in figure 1(e), both methods achieve a similar error of 1e-6. This discrepancy in performance, highlights the robustness of WNG to parameterization of the model.”
> >
> >
> > 4. Globality vs Locality: Thank you for raising this point, which is indeed essential and perhaps related to previous questions. In this sentence, we referred to equation 4, which represents the penalized objective (introduced in Pacchiano et al. (2020)). This penalized objective consists in an error term $\mathcal{L}(\theta)$ and adds an explicit term that we called the Wasserstein penalty. This penalty is global because it computes the Wassestein distance between successive updates.
> > Our proposed method does something different. It computes the gradient of $\mathcal{L}(\theta)$  and preconditions it with the inverse of the WIM (as discussed in the sentence right after equation 8).
> > As you pointed out,  equation 6 holds locally and provides a Taylor expansion of the Wasserstein, which is used to compute the WIM. This makes our approach local, compared to the global approach used in equation 4.
> >
> >
> > **Response to minor comments:** Thank you for pointing those out. We updated the paper to incorporate these changes.
> > In particular, we replaced $f$ by $f_u$ in Equation 5 as suggested. We also clarified that, in  Fig1.c,  $\sigma$ refers  to the std of the gaussian and $\mu$ to  its mean. We fixed this in the revised version.

---

> > ### Comment · AnonReviewer1 · 2020-11-21
> > **Nice revision but its significance is till not yet up to the high standard of ICLR to me**
> >
> > Thanks for the response. I have read the response and revision and I appreciate the changes the authors have made. I think overall this is nice work that engineers things to work in practice that would be interesting to a moderate number of readers.  But I think it is not yet sufficient for ICLR publication given the high standard of ICLR for the following reasons.
> >
> > * Technical novelty: All the component methods and ideas (policy constraints for RL, behavioural embedding for RL, Wasstertein gradient) are already known and well studied, and this work combines them together to make it work. The idea of using Wasserstein gradient instead of Euclidean gradient to constraint the policy update is very natural. The extension of WGN from re-parameterization to gradient log score is also very minor. Also, I don't think it is fair to compare the novelty of this work with that of TRPO: TRPO is one of the first, successful works for policy update constraints with a good guarantee and strong applicability, and different works could follow to propose different (and even better) ways for policy update constraints as in this present work.  Now I am not criticizing you on this nor saying that strong technical novelty is necessary for publication in ICLR. It is just that I will put very high weight of my evaluation on the paper's empirical significance and applicability rather than technical novelty.
> >
> > * Empirical significance and applicability:  The most important result is perhaps Figure 2. There, I compare BG-WNPG (the best version of the proposed algorithm) with its comparable method: BGPG. Out of 6 tasks there, BG-WNPG outperforms BGPG in 3 tasks (Ant, Walker2D, Inverted), has similar performance in 2 tasks (Hopper and Reacher) and underperforms  BGPG in 1 task (Halfcheetah). This result looks encouraging but does not look significant enough to me. If I am a practitioner, I am not sure if I'm convinced enough to adopt this method.

---

> > > ### Author Response · Authors · 2020-11-21
> > > **Improving the Clarity of Our Results**
> > >
> > > Thank you very much for your comments and detailed response.
> > >
> > > First, we totally agree that TRPO is a highly significant/impactful work and that is precisely what we meant in the response--we were only highlighting the relatively ‘simple’ idea of adding a KL penalty to the objective function. We apologize for any confusion on this point. We are sorry that you see our work as having low technical novelty, but we would like to address the point about empirical significance and applicability.
> > >
> > > **Empirical Significance and Applicability:**  We think that our plots in Figure 2 were too small/unclear, we would like to apologize for this as well. We have since slightly enlarged the plots and reduced line widths, making the differences between methods more clear.
> > > To further increase clarity, we’ve also added a table reporting the final performance for each PG task (Table 1 in the appendix).
> > > The table shows that our method BG-WNPG does ultimately outperform *all* on all tasks, by a fairly large margin.
> > > The table also shows that our second method, WNPG, also outperforms  *all*  baselines, including BGPG  on 5 out of 6 environments. On HalfCheetah, it achieves a similar reward to BGPG.
> > >
> > > Not only do our methods improve the final reward, but they do it faster than the baselines methods.  Figure 5 shows that our methods need fewer simulations to obtain larger rewards, which is critical in the context of RL where simulations can be expensive.
> > >
> > > We also refer to the ES experiments in Figure 4, on which our approaches also significantly outperform the baselines.
> > >
> > > From the perspective of a practitioner, we also believe a strength of our methods is that we show that the performance gap to baselines is in line with our theoretical understanding of WNG. That is, problems that are poorly conditioned result in stronger performance of WNG-based methods. From a practical standpoint, if one’s problem is ill-conditioned, we believe our results provide strong evidence that WNG-based methods will deliver (especially) robust improvements. We are also planning to release our code as soon as possible.
> > >
> > > Thank you very much once again for your time and for your comments.

---

> > > > ### Comment · AnonReviewer1 · 2020-11-25
> > > > **A good interpretation and presentation; increase my score**
> > > >
> > > > I think that's a good interpretation and presentation of the empirical significance and applicability of the proposed method. I highly recommend the authors to highlight this discussion in the final paper, especially the ill-conditioned argument, as it is highly relevant to the practitioners. I think this paper can be interesting for a moderate number of readers. I am going to increase my score to 6.

---

### Official Review · AnonReviewer2 · 2020-10-29
**Well reasoned extension of recent work to make Wasserstein NGD more scalable. Accept**

**Rating:** 8
**Confidence:** 4

**Review:**


Amari's Natural Gradient has been very successfully applied for policy optimization, e.g. in a recent line of work by Schulman et al. These benefit of using these natural gradients that restrict the change in KL divergence between successive policies was more stable and faster convergence. However two distributions may differ a lot in terms of KL divergence but because of the dynamics of the MDP they may still have almost the same behavior, therefore recent work has focused on using the Wasserstein distance to measure the divergence between successive policies which naturally leads to "Wasserstein Natural Descent". In this paper the authors build upon recent work on Kernelized Wasserstein NGD by making it more widely applicable and scalable. Moreover they present a good empirical comparison between KL-NGD and Wasserstein-NGD on a combination of pedagogical toy problems and some standard RL benchmarks from OpenAI gym. Overall this paper will be a good contribution to the conference and I recommend acceptance.


**Corrections and suggestion for improving presentation**

1. [Citations] Page 2, third paragraph from bottom cites Schulman 2015 instead of Schulman 2017 for PPO. Page 4 last paragraph has a citation missing and Li and Zhao "Wasserstein Information Matrix" paper is cited twice.

2. I think it will be better to write down that equation (5) defines $f_u$. Also shouldn't the $1/2$ factor in equation (5) be actually $\beta / 2$ ?

3. On page 3 it is said that "the penalty only accounts for global proximity in behavior .... " in reference to equation (4), but PPO is not implemented with a single value of $\beta$, i.e. the strength of the penalty typically varies as optimization goes on. Why can't the same be done for the weight of the $W_2$ penalty ?

---

> ### Author Response · Authors · 2020-11-17
> **Authors' Response**
>
> Thank you very much for the positive evaluation and suggestions for improving the paper! We are glad to read that you find the paper would be a good contribution to ICLR. We incorporated the suggestions and clarifications in the revised version as follows:
>
> 1. Thank you for pointing out those--we have adjusted the manuscript accordingly. The missing citation was the work of Tang & Agrawal (2019) on implicit policies.
>
> 2. Thank you, it is indeed clearer this way. We added the following sentence at the end of the first paragraph of section 3: “Hence, the optimal function $f_u$ solving Equation 5 defines the \textbf{optimal} vector field $\nabla_x f_{u}(x)$.”  We  could also use $\beta/2$ in equation 5. This has the effect of rescaling the WNG by $1/\beta$. Since this rescaling can be absorbed by the step-size during optimization, we preferred to keep $\beta=1$ so that we recover the WIM as defined in Li&Zhao (2019).
>
> 3. This is a very good point! Thank you for pointing this out! Indeed  $\beta$ could also be adapted in the case of W_2 penalty. We clarified this in the revised version by changing the last paragraph of section 2 and by saying:
>
> “This procedure is highly accurate when the Wasserstein distance changes slowly between successive updates, as ensured when $\beta$ is large. At the same time, larger values for $\beta$ also mean that the policy is updated using smaller steps, which can impact convergence. An optimal trade-off between speed of convergence and precision of the estimated Wasserstein distance can be achieved using an adaptive choice of $\beta$ as done in the case of PPO (Schulman et. al. 2017). For a finite value of $\beta$, the penalty accounts for \emph{global} proximity in behavior and doesn't explicitly exploit the local geometry induced by the BEM, which can further improve convergence. We introduce an efficient method that explicitly exploits the local geometry induced by the BEM through the Wasserstein Natural gradient (WNG) leading to gains in performance at a reduced computational cost.”

---

### Official Review · AnonReviewer5 · 2020-11-06
**An interesting approach to measuring policy similarity in reinforcement learning**

**Rating:** 5
**Confidence:** 2

**Review:**

The paper introduces methods for reinforcement learning based on a Wasserstein Natural Gradients (WNG), an approach to measuring similarity between policies. The methods are based on Policy Gradients and Evolutionary Strategies and add policy similarity term based on WNG instead of KL constraint as in TRPO.

The paper is well written and easy to follow (the conclusion section is missing though). Although the method is interesting, I think that the current experimental evaluation has significant flaws: the method does not demonstrate the state-of-the-art performance and significantly improves over the baselines on a small number of tasks. I believe that the paper will significantly benefit from comparisons  with stronger baselines such as PPO. Moreover, the experiments performed in the paper (figure 2) demonstrate that the approach only marginally outperforms the baselines on the harder HalfCheetah and Hopper tasks that raises concerns regarding the generality of the approach.

The paper proposed an interesting approach to policy constraints in RL but the experimental evaluation is not sufficient.

---

> ### Author Response · Authors · 2020-11-17
> **Authors' Response**
>
> Thank you for your encouraging and constructive comments. We are happy to hear that you find the method interesting and the paper well written and easy to follow. We hope the following additions and clarifications address your current concerns:
>
>
> 1. Experimental evaluation:
> Thank you for encouraging us to strengthen the empirical evaluation of the method. We performed additional policy gradient experiments on the Roboschool Ant and Reacher tasks which are now included in the revised version. The Ant task is the highest dimensional task to which we applied policy gradients. Our methods outperforms the other ones on both Ant and Reacher, which we hope further underscores the potential of our approach.
>
>
> 2. “The paper will significantly benefit from comparisons with stronger baselines such as PPO.”:
> We absolutely agree. We have now clarified in the revised version of the paper that  the results in Figure 2 already includes PPO with Clipped Surrogate Objective that was labeled as ‘KL’ and is now labeled as PPO (Clip) for clarity.  We apologize for the confusion this created. We would like to emphasize that this method was already shown to outperform TRPO and KL-penalty on various tasks in (Schulman et al. 2020). We would also like to clarify that our experimental setting is similar to the one in Pacchiano et al. (2020) and reproduces the baselines from that paper.
>
>
> 3. Improvement on the harder HalfCheetah and Hopper tasks.
> While the improvement on those tasks is indeed smaller compared to the other two tasks, we do not believe that HalfCheetah and Hopper are the hardest tasks by at least two criteria : dimension of the  observation and action spaces
> and hardness of initial improvement.
> Dimension: For instance, Walker2d has observation and action spaces that are twice as large as Hopper (and roughly the same as HalfCheetah), yet the improvement of our approach on Walker2d is significant.
> Difficulty of initial improvement:  All methods showed a fast initial improvement on both HalfCheetah and Hopper, whereas initial progress on Walker2d and InvertedDoublePendulum was more challenging. In both tasks where initial progress was difficult, the gains for WNG were most evident.
>
>
> 4. When to expect the most improvement with WNG?
> 	To investigate why WNG-based methods deliver bigger gains on some tasks compared to others, we appealed to the prior work of Arbel et al. (2020) on WNG which shows that the highest improvements are obtained when the problem is ill-conditioned. In Figure 3, we found that indeed, the problems on which WNG showed less improvement compared to baselines were those with better conditioning.
>
>
> 5. Conclusion:
> We have now included a conclusion which is as follows:
> “Explicit regularization using divergence measures between policy representations has been a common theme in recent work on policy optimization for RL. While prior works have previously focused on the KL divergence, Pacchiano et al. (2020) showed that a Wasserstein regularizer over behavioral distributions provided a powerful alternative framework. Both approaches implicitly define a form of natural gradient, depending on which divergence measure is chosen. Through the introduction of WNPG and WNES, we demonstrate that directly estimating the natural gradient of the un-regularized objective can deliver greater performance at lower computational cost. These algorithms represent novel extensions of previous work on the WNG to problems where the reparameterization trick is not available, as well as to black-box methods like ES. Moreover, using the WNG in conjunction with a WD penalty allows such penalty to take advantage of the local geometry induced by WNG, further improving performance. We also provide a novel comparison between the WNG and FNG, showing that the former has significant advantages on certain problems. We believe this framework opens up a number of avenues for future work. Developing a principled way to identify useful behavioral embeddings for a given RL task would allow to get the highest benefit form WNPG and WNES. From a theoretical perspective, it would be useful to characterize convergence boost granted by the combination of explicit regularization and the corresponding natural gradient approach.”
>
>
>
> (Schulman et al. 2017):  Proximal Policy Optimization Algorithms, https://arxiv.org/abs/1707.06347
> (Pacchiano et al. 2020): Learning to Score Behaviors for Guided Policy Optimization,  https://proceedings.icml.cc/static/paper_files/icml/2020/2630-Paper.pdf

---

### Public Comment · ~Wu_Lin2 · 2020-11-10
**Some questions**

Interesting work. I have some questions.

According to Table 1 of  Li & Zhao (2019b), the *exact* Wasserstein information matrix in Gaussian cases is the identity matrix.  Since the Wasserstein Natural Gradient depends on the Wasserstein information matrix, in Gaussian cases (see Figure 1 of this ICLR submission),  Wasserstein natural gradient descent becomes SGD if the exact Wasserstein information matrix is used.
Figure 1 shows that Wasserstein natural gradient descent, in this case, SGD performs better than the Fisher natural gradient descent, which looks surprising.
Q1:  I wonder if the exact Wasserstein information matrix and the exact Fisher information matrix are used in Figure 1 since they have a closed-form expression and can be efficiently computed in this case.
In other words, do the authors use any Monte Carlo approximation to approximate both information matrices in Figure 1?
If a kernel trick is used to approximate the Wasserstein information matrix, is it possible that the kernel is more powerful than the q distribution? For example, a Gaussian/RBF kernel can capture a high-order correlation between two particles while a diagonal Gaussian q only can capture the second-order correlation due to the diagonal covariance parameter.  This could imply that the usefulness of Wasserstein natural gradients depends on the choice of a kernel and its hyper-parameters.  According to <1>, I wonder whether the  Wasserstein natural gradient method suffers from the curse of dimensionality.

In practice (both in RL and variational inference), we usually maximize E_q [ \psi(x) ]  + E_q [ - \log q] instead of E_q [ \psi(x) ], where the entropy term is intrdouced. In Gaussian cases, the entropy term does not affect the optimal solution and is useful for exploration in RL.
Q2: I wonder if the entropy term plays a role in Figure 1 or not. In other words, if the entropy term is included,  do we have the same observation as shown in  Figure 1?
Q3: Is the entropy term included in the evolution strategy tasks?

Reference
<1> Ba, Jimmy, et al. "Towards Characterizing the High-dimensional Bias of Kernel-based Particle Inference Algorithms." (2019).

---

> ### Author Response · Authors · 2020-11-15
> **Thank you for the interesting questions!**
>
> Hi Wu Lin,
>
> Thank you for your interest in our work! Those are very good questions! We provide the following answers:
>
> # Exact WIM of a Gaussian:
> Thank you for raising this point, it is an important question in the context of WNG. We  provide below the exact expressions of the WNG used in Figure 1 and as you will see they are indeed different from the euclidean gradient used in SGD. As you pointed out, the table 1 of  Li & Zhao (2019b)   states that the WIM of a  **particular parameterization** of the  gaussian is the identity, thus the WNG descent becomes SGD.  In their case, the gaussian is parameterized by the mean and standard deviation.  However, using a different parameterization of the same gaussian, results in a different expression for the WIM and thus for the Wasserstein natural gradient. In figure 1, we used two parameterization, neither of which results in the identity for WIM.  For instance, if the gaussian is parameterized by its covariance matrix (instead of std), this results in an expression for the natural gradient that is different from the euclidean gradient. (see [1] Equation 101, page 17).
>
> [1] L. Malag`o, L. Montrucchio, and G. Pistone. Wasserstein Riemannian Geometry of Positive Definite Matrices. arXiv:1801.09269 [math, stat], Jan. 2018. arXiv: 1801.09269.
>
>
> ## Expressions of the WNG and FNG:
> We report now the expression we used in figure 1:
> ### diagonal parameterization
> In this case we used  $\Sigma = diag(v) $,  the parameters are $\theta = (\mu, v  )$.
> #### WNG:
> The gradient wrt to the mean parameter remains the same as the usual euclidean gradient. However, the gradient wrt to $v$ is given by:
> $ \nabla_{v}^{W} \mathcal{L}(\theta ) =  4*\Sigma \nabla_{v} \mathcal{L}(\theta),  \qquad    \nabla_{\mu}^{W} \mathcal{L}(\theta ) =  \nabla_{\mu} \mathcal{L}(\theta )   $
> #### FNG:
> Those are given by:
> $ \nabla_{v}^{F} \mathcal{L}(\theta ) =  2 \Sigma^2\nabla_{v} \mathcal{L}(\theta), \qquad   \nabla_{\mu}^{F} \mathcal{L}(\theta ) =  \Sigma\nabla_{\mu} \mathcal{L}(\theta)$.
>
> ### log-diagonal parameterization
> In this case we used  $\Sigma = diag(\exp(2v)) $,  the parameters are $\theta = (\mu, v)$.
> #### WNG:
> Again, the gradient wrt to the mean parameter remains the same as the usual euclidean gradient. However, the gradient wrt to $v$ is given by:
> $  \nabla_{v}^{W} \mathcal{L}(\theta ) = \Sigma^{-1} \nabla_{v} \mathcal{L}(\theta ),\qquad   \nabla_{\mu}^{W} \mathcal{L}(\theta ) =  \nabla_{\mu} \mathcal{L}(\theta )$
> #### FNG:
> Those are given by:
> $ \nabla_{v}^{F} \mathcal{L}(\theta ) =  .5*\nabla_{v} \mathcal{L}(\theta), \qquad \nabla_{\mu}^{F} \mathcal{L}(\theta ) =  \Sigma\nabla_{\mu} \mathcal{L}(\theta)$
>
>
> ### Summary:
> As we see, the WNG wrt to the mean parameter is always equal to the euclidean gradient. However, the WNG wrt the parameters of the covariance depends on the parameterization.
> For FNG, in both parameterizations, the FNG wrt to the mean parameters is obtained from the euclidean one by conditioning it by the covariance matrix. In our example, since the optimal solution is  obtained for $\Sigma = 0$ and $\mu=0$, this means that FNG slows down the update of $\mu$ because they are being multiplied by $\Sigma$ which gets closer to $0$. This doesn’t happen in the case of WNG. While this might look surprising at first, we believe the closed form expression of these gradients provides a reasonable explanation.

---

> > ### Author Response · Authors · 2020-11-15
> > **Rest of the response 2/2**
> >
> > ### 2- Did we use Monte Carlo approximation to approximate the information matrices for figure 1?
> > No we didn’t, we used the closed form expressions for both WNG and FNG given in the above equations. Monte Carlo estimation is only used to estimate the euclidean gradients  $\nabla_{\mu} \mathcal{L}(\theta)$ and $\nabla_{v} \mathcal{L}(\theta)$.
> >
> > ### Is it possible that the kernel is more powerful than the q distribution?
> > In figure 1 we did not use any kernel trick, we directly used the closed form expressions for the WNG and FNG that are available for gaussians as described above.
> >
> > ### According to <1>, I wonder whether the Wasserstein natural gradient method suffers from the curse of dimensionality.
> > This is an interesting question which we believe was extensively addressed in Arbel et. al. 2020. At a high level we can say the following:
> >
> > In general, the **no-free lunch theorem** holds: even regression can suffer from the curse of dimensionality in the most general cases if no additional structure is available. However, the models considered here are smooth parametric models. One can therefore exploit such smoothness and come up with good estimators of WNG  that enjoy good convergence rates.
> >
> > ### Q2: I wonder if the entropy term plays a role in Figure 1 or not. In other words, if the entropy term is included, do we have the same observation as shown in Figure 1?
> > As you pointed out the entropy encourages exploration. This also means that the entropy term prevents the variance $\Sigma$ of the gaussian from going to $0$ which is precisely the optimal solution in our setting. Since the entropy goes to $+\infty$ as $\Sigma$ gets closer to $0$, we can safely guess that it would further slow down convergence in this setting.
> >
> > We hope we answered all your questions and we are happy to provide more answers.

---

### Public Comment · ~Wu_Lin2 · 2020-11-16
**Comparison between the exact WNGD and FNGD in Gaussian cases**

I thank the authors for the answer to my questions.
According to your answer to question (1), it seems that the performance of WNGD (Wasserstein Natural Gradient Descent) depends on the choice of parameterization.  WNGD becomes SGD if the parameterization in  Li & Zhao (2019b) is used. I wonder about the performance of WNGD in Figure (1) if the parameterization in  Li & Zhao (2019b) is used.


Compared to WNGD, the advantages of FNGD in Gaussian cases are:

* (1) FNGD is less sensitive to the choice of parameterization.
In <2>, a Cholesky factor of the covariance is used.
In <3>, the covariance is used. In <4>, the natural parameterization of Gaussian is used. In <5>, the precision matrix (the inverse of the covariance) is used. Under these parameterizations, FNGD seems to perform well according to these works, which implies that FNGD is invariant/robust under the choice of parameterization (see <9>). On the other hand, it seems that WNGD depends on the choice of parameterization.

* (2) Under many parameterizations, the FNGD update in the mean is $\nabla_\mu^F = \Sigma \nabla_\mu {\cal L}$  as shown in your answer.
On one hand, the authors exploit its weakness as shown in Figure 1 since in deterministic cases (optimization without the entropy term),  $\Sigma$ becomes $0 $. However, this weakness of FNGD disappears in variational inference and possibly in RL with the entropy term. Recall that in Gaussian cases, the entropy term does not change the optimal solution.
On the other hand, the FNGD update in the mean has been exploited in many papers to design adaptive gradient methods (see <5-7>), which also justify the faster convergence of FNGD at the beginning as shown in Figure 1, where stochasticity plays a role (see <8>).
In multivariate Gaussian cases with full covariance, the FNGD update in the mean is like a Newton update (see Eq (8-9) in <5> for the detail)  $\nabla_\mu^F = \Sigma \nabla_\mu {\cal L}$ while WNGD seems to be the Euclidean gradient descent  $\nabla_\mu^W = \nabla_\mu {\cal L}$.
In practice, I wonder whether the difference between error $10^{-4}$ (FNGD) and error $10^{-8}$ (WNGD) matters since we do not know the exact solution in real-world applications

Nevertheless, this work is interesting and worth for further exploration. Additional experiments about the comparison between FNGD and WNGD may be needed. It is likely that inexact WNGD is better. Even in diagonal Gaussian cases, inexact WNGD could be better than the exact WNGD and exact FNGD since a powerful kernel can capture extra information in an objective function than a diagonal Gaussian q distribution.


References
* <1> Ba, Jimmy, et al. "Towards Characterizing the High-dimensional Bias of Kernel-based Particle Inference Algorithms." (2019).
* <2> Salimbeni, H.,  et al. Natural gradients in practice: Non-conjugate variational inference in Gaussian process models.  (2018)
* <3> Tran, Minh-Ngoc,  et al. "Variational Bayes on Manifolds." (2019).
* <4> Khan, et al. "Conjugate-computation variational inference: Converting variational inference in non-conjugate models to inferences in conjugate models." (2017).
* <5> Lin, et al. "Handling the Positive-Definite Constraint in the Bayesian Learning Rule" (2020)
* <6> Khan, et al. "Fast and scalable Bayesian deep learning by weight-perturbation in adam". (2018)
* <7> Zhang, et al. "Noisy Natural Gradient as Variational Inference" (2018)
* <8> Amari, et al. "When Does Preconditioning Help or Hurt Generalization?" (2020)
* <9> Song et al. "Accelerating natural gradient with higher-order invariance." (2018).


 Summary:
* (1) This work is interesting and worth for further exploration.

* (2) The performance of exact WNGD may depend on the choice of parameterization.

* (3) It is likely that inexact WNGD is better than exact WNGD since a kernel can be used in inexact WNGD and the kernel may be able to capture extra information than the q distribution does. Even in diagonal Gaussian cases, inexact WNGD could be better than exact WNGD and exact FNGD due to the use of a powerful kernel.

* (4) Additional comparison studies between exact FNGD and exact WNGD may be needed.
On one hand, the authors exploit the weakness of FNGD in deterministic cases (optimization without the entropy term). However, this weakness of FNGD disappears in variational inference and possibly in RL with the entropy term.  On the other hand, the FNGD update has been exploited in many papers to design adaptive gradient methods. This is the reason why FNGD converges faster than WNGD at the beginning, where stochasticity plays a role.
In multivariate Gaussian cases with full covariance, the FNGD update in the mean is like a Newton update while WNGD seems to be the Euclidean gradient descent.

* (5) It is interesting to see a comparison study among exact WNGD, inexact WNGD, and exact FNGD in Gaussian cases with full covariance (without considering computation cost) in a 100-dimensional correlated setting.

---

> ### Author Response · Authors · 2020-11-20
> **Thank you for your comments**
>
> Dear Wu Lin,
>
> Thank you for your comments. Those are interesting considerations. Here are a few comments  about the points in your summary:
> #### (1)- Thank you!
> #### (2)- Invariance to parameterization:
>  Invariance to parameterization for WNG is a known fact resulting from differential geometry: please refer to [1,2,3] and  to Proposition 1 in [4]. The same holds for FNG. The expressions automatically adapt to the choice of parameterization so that the trajectories in probability space remain invariant.
> - Note that all the parameterizations we considered in Figure 1  and including the one you suggested are for the covariance matrix and not for the mean. Thus only the gradient of the covariance parameter is affected and not the mean. Hence, the  FNG wrt to the mean vector will still be multiplied by a shrinking covariance, which is responsible for the slowing down of FNG.
> - Please note that the parameterization you suggest is a particular choice for which SGD happens to match WNG, since trajectories of SGD depend on parametrization. This particular choice benefits from the advantages of WNG which gives an improved performance.
>  Figure 1 (e) already shows similar performance between WNG and SGD. This doesn’t hold true in the case of Figure 1 (d) where WNG improves over SGD.
> - Thus, the advantage of WNG is that it maintains good performance regardless of the choice of parameterization as shown by Figure 1 (d,e).
>
> #### (3)- Approximating WNG using kernels:
> Please refer to Theorem 7 in [4] which shows that approximating WNG using kernels gets closer to the exact WNG as the number of samples increases and quantifies the speed of convergence.
> #### (4)-  Variational Inference:
> FNG is certainly a powerful tool in variational inference for its connection to KL and entropy.
> In this work, we consider problems that are different from variational inference and for which other conclusions might hold.
> Figure 1 simply illustrates that, for problems outside of the framework of variational inference, it can also be worth considering other tools, such as WNG.
> ##### (*) Gauss-Newton:
> - FNG recovers an update similar to Gauss-Newton for a particular choice of loss: the log-likelihood of a multivariate gaussian.
> - WNG also recovers a Gauss-Newton update for a different choice of loss: the Wasserstein distance.
>
> #### (5)- Full gaussian:
> Similar expressions hold for WNG and FNG, thus the simpler diagonal example already captures the essence of what is happening. We refer to our experiment section 6 which uses the estimator to solve more challenging tasks.
>
>
> [1]  L. Malag`o, L. Montrucchio, and G. Pistone. Wasserstein Riemannian geometry of Gaussian densities. Information Geometry.
>
> [2] W. Li and G. Montufar. Natural gradient via optimal transport.
>
> [3] W. Li and J. Zhao. Wasserstein information matrix.
>
> [4] M. Arbel, A. Gretton, W. Li, and G. Montufar. Kernelized Wasserstein Natural Gradient.
>
> Thank you once again for your interest in our work, and we hope that we've addressed your concerns!

---

### Author Response · Authors · 2020-11-17
**General Response to All Reviewers**

Thank you all very much for your useful insightful comments and suggestions! We very much appreciate that you all see our approach as interesting. We have updated a new revised version of the document with the following additions:

1- Additional experiments:
As suggested by AnonReviewer1 and  AnonReviewer5, we added more experiments on two new tasks from the Roboschool suite of experiments, Ant and Reacher. Our methods deliver strong performance on both.

2- Comparison with PPO:
Following the remark of  AnonReviewer5,  we have clarified in the document that one of the methods we compared against and labeled as ‘KL’ refers to PPO with Clipped Surrogate Objective (Schulman 2017). We apologize for the confusion this might have created.
We also highlight that our method improves over PPO which in turn was already shown to outperform TRPO on various tasks in Schulman et al. (2017).

3- Conclusion:
Following the remark of AnonReviewer5, we included a conclusion to discuss the contribution and possible future work.

4- Experimental setting of Figure 1:
We included the experimental details to produce the results of Figure 1. In particular, we provided the closed-form expressions of WNG and FNG in the Appendix D.3.

5- Further clarifications:
We also included further clarifications/ corrections in the main text as suggested by the reviewers. Each of those modifications are discussed in the individual responses.


Thank you all very much for your time, and we hope these changes address your concerns.

---

### Decision · Program_Chairs · 2021-01-07
**Final Decision**

**Decision:**

Accept (Poster)

**Comment:**

This paper explores the Wasserstein natural gradient in the context of reinforcement learning. R5 rated the paper marginally below the acceptance threshold, but is not very confident about the correctness of his/her assessment. His/her main criticism was the experimental evaluation. This concern was shared by a confident R1. R1 found the paper well structured and that it contains encouraging empirical results, but low technical novelty and (initially) insufficient experiments. His/her initial recommendation was reject, but following an extensive discussion and improvements of the manuscript by the authors, he/she was more convinced about the empirical significance and applicability of the method, and raised his/her score to 6, indicating that the interpretation and presentation improved but that the paper might be interesting only to a moderate number of readers. A confident R2 found this paper very good, although only providing a short review. Two other unfinished or not sufficiently confident reports were not taken into account. Weighing the reports by contents, confidence, and participation in the discussion, the paper scores marginally above the acceptance threshold. In view of the authors' responses, I am discounting R5's criticism about lack of comparison with the PPO baseline. I personally consider the paper very well written, that it presents a natural and potentially useful application of the Wasserstein natural gradient to the context of reinforcement learning, and enjoyed the discussion of behavioral geometry. I am recommending a borderline accept. However, I also appreciate the concern of the referees about the limited technical innovation and how some of the strengths of the method could be presented more convincingly. Please take these comments carefully into consideration when preparing the final version of the paper.